# Emerging Insulin Analogues: A Glimpse into How Insulin Analogues May Look in the near Future

**DOI:** 10.3390/pharmaceutics17101239

**Published:** 2025-09-23

**Authors:** Ntethelelo Sibiya, Lorah Dzimwasha, Samarah Zvandasara, Amanda Zuma, Andile Khathi

**Affiliations:** 1Pharmacology Division, Faculty of Pharmacy, Rhodes University, Makhanda 6139, South Africa; lorahdzimwasha@icloud.com (L.D.); teamsamarah@gmail.com (S.Z.); a.zuma@ru.ac.za (A.Z.); 2School of Laboratory Medicine and Medical Sciences, University of KwaZulu-Natal, Durban 4001, South Africa; khathia@ukzn.ac.za

**Keywords:** insulin analogues, thermostability, diabetes, stability, hepato-preferentiality

## Abstract

The use of insulin as a treatment for diabetes mellitus has been marred by several challenges. These setbacks incurred in an attempt to better manage diabetes, together with past innovative strategies, have encouraged science and the clinical community to continue to endeavour for an ideal insulin analogue that demonstrates heightened pharmacokinetic profiles and thermal stability. This review therefore seeks to provide an update on emerging insulin analogues. This review aims to update the science and clinical community of the recent developments on novel insulin analogues design and engineering. Through this exercise, we believe this review consolidates cutting-edge innovations on insulin development and diabetes management. The accelerated innovation of design and engineering in the biotechnology and peptide field has seen more insulin analogues reported in the last decade. Although the analogues are often limited to preclinical studies, Degludec, Icodec, and Efsitora have been the exceptions. The emerging insulin analogues include those with extended pharmacokinetic profile, increased thermostability, are glucose-responsive, and are hepato-preferential insulin analogues. Due to the fast pace of innovation in the design of insulin analogues, more insulin analogues are likely to emerge in the clinical space in the near future. Such innovations should be applauded and encouraged as they aim to strive for better management of diabetes mellitus.

## 1. Introduction

Over the past few decades, insulin has contributed immensely to the treatment of diabetes mellitus. The role of insulin in glucose metabolism has been well established, leading to the development of several insulin analogues. To improve glycaemic control, scientists have developed short-acting, intermediate-acting, and long-acting insulin analogues [1]. Insulin is used in the treatment of type 1 diabetes mellitus and late-stage type 2 diabetes mellitus. This peptide is typically administered as a subcutaneous injection once or twice daily [2,3].

Research evidence shows that insulin intensive therapy effectively prevents the onset of microvascular complications; however, the burden associated with multiple injections tends to diminish the patient’s compliance, thus translating to poor glycaemic control [4,5]. To try and circumvent this challenge, scientists aimed to develop insulin analogues with extended pharmacokinetic profiles, which can minimise the frequency of injections whilst affording adequate glycaemic control. These strides have resulted in the development of insulin Icodec and Efsitora, insulin analogues injected once weekly.

The intrinsic instability of the insulin peptide is a critical obstacle faced during formulation because it is susceptible to aggregation and fibrillation, especially under stressful conditions such as low pH or high temperature. These conditions cause insulin to misfold and aggregate, initially forming short protofibrils that eventually mature into longer insulin fibrils [6,7]. Although essential, the need for refrigeration may impose a financial burden on patients in areas with limited resources. Without stability in insulin formulation, patients are exposed to fibrils that have long-term harmful effects. In addition, formulations do not have a sustainable shelf life, subsequently increasing manufacturing costs and the burden of that cost falls onto the patient. Such formulation failures then contribute to non-adherence to medication in patients and increased prevalence of uncontrolled glycaemia. As a result, researchers have proposed the development of ultra-stable insulin analogues capable of resisting fibrillation.

A chief side effect of insulin therapy is hypoglycaemia. Several steps have been taken to address this issue, including developing insulin analogues with predictable pharmacokinetic and pharmacodynamic properties. About half a century ago, Brownlee and Cerami proposed the concept of a glucose-responsive insulin to help mitigate the insulin-induced hypoglycaemia [5]. The idea inspired many designs intending to prevent hypoglycaemia by adjusting the release of exogenous insulin release in response to blood glucose concentrations. [4].

The natural insulin gradient, which prioritises delivery to the liver over other organs, is not honoured by subcutaneous insulin administration [2,6]. A function of insulin in hepatic cells is the inhibition of gluconeogenesis which promotes glycogen storage and production and further secretion of glucose in the blood. The absence of insulin hepatic cells contributes to hyperglycaemia in patients with diabetes despite subcutaneous administration of insulin due to its poor circulation to the liver.

A hepato-preferential insulin analogue has been developed to restore this physiological balance, and this article will discuss its effectiveness. We believe that because of the rising global prevalence of diabetes mellitus, a commentary on emerging insulin analogues as treatment options is important for the medical community. In this article, we discuss and summarise cutting-edge innovation in the design of these new insulin analogues and their effects.

## 2. Methods of Literature Search

A systematic literature search was undertaken to adequately explore and assemble relevant information pertaining to the discovery and establishment of emerging insulin analogues. The scope of this article includes, but is not limited to, the development of novel insulin analogues with enhanced thermostability of insulin, a vital component for the treatment of diabetes mellitus which is a prevalent condition plaguing the country and the globe. Esteemed scientific databases such as Google Scholar, PubMed, and Scopus, were searched and utilised in the extraction of data from English written peer-reviewed journal articles published within the past 10 years in order to preserve the relevance of the study. This method of formulating the literature review established evidence-based research.

## 3. General Physiology of Insulin

Insulin is synthesised by β cells of the Islet of Langerhans within the pancreas, producing a precursor protein known as proinsulin within the endoplasmic reticulum (ER) [8,9]. The formation of the inter- and intrachain bonds follows the first stage of the maturation process consisting of insulin cleaving of the first domain by endopeptidases, producing proinsulin. Proinsulin then exits the ER via exocytosis and is translocated to the Golgi apparatus, where matured insulin is produced [10,11,12]. Insulin is secreted as hexamers, formed by the chelation of insulin monomers with zinc ions [13]. In blood plasma, the propensity for self-assembly weakens, thus releasing monomeric insulin forms, the only bioactive form. This phenomenon has been central in developing insulin forms with altered pharmacokinetics and pharmacodynamic profiles.

### 3.1. Insulin Signalling

Insulin regulates many pathways at molecular levels, such as lipid synthesis and storage, stimulation of protein synthesis, inhibition of ketogenesis and gluconeogenesis, stimulation of glycolysis and glucose storage, and cell proliferation [14]. Once insulin is released into the circulation, insulin forms an insulin–receptor complex with the tyrosine kinase receptor, culminating in the translocation of glucose transporter (GLUT4) which facilitates glucose uptake [15,16,17,18,19]. The translocation of GLUT 4 transported is stimulated by the cascade of activation reactions that occur. From the activation of the tyrosine kinase receptor which activates intracellular signalling substrates, which go on to induce the embedding of GLUT 4 transporters into the cell membrane for glucose uptake. The process described here is the insulin transduction pathway, depicted in Figure 1, illustrating the insulin’s potential in mitigating the mechanisms that lead to the manifestation of diabetes mellitus.

Insulin has a very short half-life (4–6 min). Apart from the first hepatic metabolic clearance, which sees approximately 80% of secreted insulin degraded, insulin can also be metabolised by the peripheral tissues including the kidneys [20,21].

### 3.2. Insulin’s Architecture

Insulin is a relatively small protein with a molecular mass of 2808 Daltons. The primary structure of this peptide, depicted in Figure 2A, comprises two peptide chains: the A chain, which consists of 21 acidic amino acids and the other, the B chain, which has 30 basic amino acid residues. At the tail end of the B chain, a high quantity of free α-amino acids (phenylalanine) makes up that chain’s N-terminal. The two peptide chains are bound by two separate disulphide bonds, forming an interchain bridge between the cysteine residues, found on both chains [22,23,24].

Insulin’s secondary structure is illustrated in Figure 2B by forming α helices antiparallel to each other along the amino acids A^2^–A^8^ and A^13^–A^20^ within the A chain [25,26,27,28]. The possession of polar and non-polar amino acid residues implies that the shell of the hormone is susceptible to agglomeration with similar insulin monomers. The extent of aggregation of insulin monomers depends on the concentration of the monomers and zinc ions (Zn^2+^) plus the pH of the solution. As illustrated in Figure 2C, monomers of insulin tend to form dimers first; with increased concentration, the monomers oligomerise further to form tetramers, and finally, in ideal conditions such as a pH level of around 6 and Zn^2+^ ion concentration of 10mM, the insulin forms hexamers [29,30].

### 3.3. Conventional Insulin Analogues

Human insulin was the first marketed form of endogenous insulin in the 1920s [31,32]. The development of insulin has evolved, demanding diversified analogues of insulin that have different action times and are administered in manners that mimic the body’s pulsative secretion of endogenous insulin. The understanding of the insulin molecule, including its isoelectric character, solubility, and receptor binding has allowed the design of insulin analogues with varied pharmacokinetics [33,34,35,36]. Currently, three categories of insulin are available: rapid/short, intermediate, and long-acting insulin analogues.

Short or rapid-acting insulin analogues, such as insulin Lispro (Figure 3), are designed by a paired amino acid substitution of proline and lysine at positions B^28^ and B^29^, respectively [37]. The substitution of the two amino acids produces insulin monomers that are less likely to self-assemble, thus shortening the time for absorption and duration of action of the analogue [38]. Insulin Lispro is equipotent to human insulin in terms of biochemical signalling and insulin receptor binding [39]. Similar pharmacodynamic properties of Lispro apply to insulin Glulisine. Glulisine is the most recent insulin analogue developed with faster absorption and onset of action compared to human insulin, with a shorter duration of action shown in Figure 3. Properties of Glulisine are all attributed to the substitutions of amino acids on its B chain: lysine at B^3^ and glutamic acid at B^29^.

On Glargine, two arginine residues were added to the tail end of the B chain at B^31^ and B^32^ (Figure 3). The two arginine residues shifted the isoelectric point by +2 [40]. Glargine is formulated at pH 4 with the inclusion of Zn^2+^ ions and the exclusion of a buffer. The basis behind excluding a buffer is to allow a depot of Glargine molecules to form under the patient’s skin after subcutaneous administration when the molecules precipitate out of the solution from exposure to the physiological body pH of 7.4 [41].

Insulin detemir (Figure 3) is an analogue formulated at pH 7 and consists of hexamers [42]. Moreover, the amino acid threonine at B^30^ is removed, and the amino acid at B^29^ is acylated to improve the stability of the analogues after subcutaneous administration and protein binding to albumin. The modifications instituted for detemir formulation prolong the duration of action, absorption rate, and dosing intervals [43]. Further development of detemir gave rise to ultralong acting insulin (Degludec) which was approved in September 2015. Degludec consists of hexadecanedioic acid conjugated to lysine on B^29^, which allows the formation of multihexamers to form at the subcutaneous injection site that cannot readily be absorbed into circulation. This design has further reinforced the extended half-life of this insulin analogue, giving it an ultra-acting status that presents a peakless pharmacokinetic profile. Degludec utilises phenolic and zinc agents to stabilise post-subcutaneous injection of the preparation [44]. Despite such significant advancements, continuous innovation in diabetes management aims to attenuate hypoglycaemia risks further, reduce injection frequency, and abolish the need for cold storage.

## 4. Emerging Insulin Analogues

In the preceding section, we will briefly explore emerging insulin analogues in the following categories: ultra-long-acting insulins, thermostable insulin, glucose-responsive insulin, and hepato-preferential insulins (Table 1).

### 4.1. Ultra-Acting Insulin Analogues

The emergence of Degludec in a clinical setting has paved the way for the exploration and development of more ultra-acting insulin [45,46]. One possible way of further achieving long-lasting insulin is through the design of insulin analogues which can resist degradation by insulin degrading enzymes (IDE).

#### 4.1.1. Seleno-Insulin

In 2023, the Iwaoko group reported on a design of a seleno-insulin analogue, demonstrating a long-lasting effect potential [47]. The underlying design strategy involved replacing cysteine residues in the solvent-exposed disulphide bridge with selenium residues to form a diselenide bridge at A^7^–B^7^, shown in Figure 4. Previous studies have reported that the selenide bond offers more advantages over its disulphide counterpart. Firstly, the selenide bond has a higher rotational barrier and high redox potential because of its pKa of 5.2. The pKa allows for the deprotonation of the side chains on amino acids at physiological pH [48]. This subsequently permits folding to occur within the chain, even in the absence of extra cysteine residues.

In this study, the seleno-alpha and seleno-beta chains were synthesised using an FMOC-based solid phase peptide synthesis. Interestingly, the similarity in seleno-peptide insulin structure was observed when compared to native insulin which was supported by retention of insulin activity. The seleno-substituted insulin activated essential insulin signalling pathway proteins, including Akt and GS3K in the Hela cells at 1 µm.

The introduction of folding within the chain, even in the absence of extra cysteine residues, abolishes the need for stabilisers in the formulation and eases formulation stability during the manufacturing processes. What this suggests is a possible formulation that could have a long shelf life which is ideal considering degradation is minimised substantially. This necessitates the possibilities for government subsidising and expanding access to the insulin therapy that aids in the overall control of glycaemia in patients with diabetes and therefore plays into reducing the prevalence of diabetes.

Despite these observations, perhaps more studies focused on the exploration of primary insulin sensitive cell such as the differentiated skeletal muscle and adipose tissue, could provide robust data on the effect of such analogues on key transporters involved in GLUT 4 transport, therefore supporting its effect. Most importantly, the authors discovered a remarked resistance to degradation by IDE, where seleno-insulin structural stability was retained for 8 h compared to 1 h achieved by the wild-type insulin. A resistance to the degradation enzymes would typically cause an increase in the residence time of the seleno-insulin. Therefore, the increased residence time would permit an increase in the biological response of the IR for the stimulation of the insulin signal transduction pathway However, when administered in animals, the seleno-insulin demonstrated a pharmacodynamic effect mirroring a short-acting insulin, whilst being able to sustain a glucose-lowering effect for 24 h. Despite these promising observations, comparative studies with conventional long-acting insulin such as glargine could be pivotal in facilitating the extended pharmacokinetic profile of seleno-insulins. Furthermore, considering the chronic nature of the disease, long-term studies are required. These studies could be instrumental in understanding the hypoglycaemic profile of seleno-insulin analogues together with their potential toxicological effect, for example, cardiovascular, hepatic, and renal hazards. From a manufacturing point of view, FMOC-based protein synthesis methods could be associated with high cost. Perhaps, biotechnological studies, including recombinant strategies such as using *E.coli* and yeast expression systems, contributing to the development of seleno-insulin presents a unique strategy to develop long-acting insulin.

#### 4.1.2. Degludec-like Insulins

Pan and colleagues engineered insulin analogues with variations at the B^36^–B^30^ region, there the hexadecanedioic acid was appended to the end of Lys B^29^, together with various amino substitutions along the chain, as shown in Figure 5 [49]. The novel insulin analogue, which retained receptor binding affinity, had AspB^28^ and GluB^30^ substituted with ProB^28^ and ThrB^30^. It is this analogue that was selected for further in vivo studies. This potential analogue has since been named INS061. Substituting acidic with neutral amino acids was sought to increase the oligomerisation into hexamers, which was observed in long-acting insulin. INS061 sustained a blood glucose-lowering effect for a 24 h duration in streptozotocin-induced diabetic rats, which was superior to Degludec, a structurally similar conventional analogue. The glucose-lowering effects peaked between 4 and 8 h, revealing better AUC results than Degludec [49].

Furthermore, the subcutaneous administration of INS601 indicated a rapid absorption phase, with a T_max_ of between 2.33 and 3.5 h. In addition to the extended half-life, INS601 exhibited an absolute bioavailability of 70.04% and a half-life of 6.06–6.53 h. INS601, based on our knowledge, is the latest emerging analogue which has undergone pharmacokinetics, distribution, and elimination studies in a preclinical stage. Over and above that, INS601 shows a comparable half-life over Degludec in rats. The observations from this study hold promise on the druggability of INS601, pending safety profiles and warranting further development.

### 4.2. Thermostable Insulin

The intrinsic instability of the insulin peptide makes it susceptible to aggregation and fibrillation, especially under stressful conditions such as low pH or high temperature. This can lead to a reduction in insulin availability and poor glycaemic control which is a challenge for many patients [7]. Even in the presence of 3 disulphide bonds and insulin’s propensity to self-assemble into hexamers, the formation of insulin fibrils is inevitable, hence the need for refrigeration [50,51]. Strategies aiming to improve insulin thermostability has resulted in the development of the following categories of insulin analogues: additional disulphide bonds, carbon-based linkers, and single-chain insulins.

#### 4.2.1. Additional Disulphide Bonds

One of the strategies being explored to attain insulin stability is the concept of introducing a fourth disulphide bond. The DiMarch group reported an introduction of the additional disulphide bond between A^10^–B^4^, where structural elucidation suggested a minor impact on the tertiary structures. Indeed, the analogue proved to have similar binding affinities as insulin receptors, just like the native insulin [52,53]. However, the pharmacodynamic response was compromised despite its improved stability.

In 2020, Xiong and colleagues further reported on four novel insulin analogues with an additional disulphide bond [54]. A different approach was sought, where the fourth disulphide bond was introduced in an extended C terminal within the A chain. Here, the A chain was extended to have 24 amino acids residues and the B chain truncated to contain 22 amino acid residues. The group successfully assembled four analogues, the differences between them being the position of the additional disulphide bond on each analogue: A^21^–B^22^, A^22^–B^22^, A^22^–B^23^, and A^22^–B^21^ with a yield of 6.4, 4.9, 9.1, and 11%, respectively. Out of the four analogues produced, only the analogue with the additional disulphide bond at position A^22^–B^22^ retained activity, as demonstrated in Figure 6. This was evidenced by its ability to activate AKT and lower blood glucose concentrations.

Although this data was promising, long-term studies are warranted, with a goal to elucidate the glycaemic control profile and potential toxicological effect. This analogue proved to be stable using a stress ageing assay, where aggregates formed only at 42 h instead of the 10 h achieved by the native insulin. Despite these developments, there have been no stability reports in relation to storage over time, and the pharmacokinetic data has not been reported. Therefore, further developments should prioritise the understanding of the effect of long-term storage at various temperatures.

#### 4.2.2. Carbon-Based Linkers

The researchers also reported on an insulin analogue where a methylene thioacetal bond replaces the intrachain disulphide bond [55]. The premise for this approach was based on providing more stability to prevent A chain folding. Reports suggest that unfolding A chain’s N-terminal helix exposes a hydrophobic surface, leading to accelerated insulin fibrillation [50,51,56]. For these reasons, replacing the A^6^–A^11^ disulphide bond with the non-reducible methylene thioacetal could stabilise the N-terminus of the A chain, leading to reduced insulin fibrillation (Figure 7). Indeed, replacing the disulphide bond with the methylene thioacetal linker gives the peptide an improved thermostability over native insulin when incubated at 60 °C for four days [55]. These observations were supported by a stress ageing assay, in which no insulin aggregation occurred before reaching a 100 h incubation period. The alteration in the intrachain bond reduced the biological activity of insulin, as the analogue showed a reduced ability to stimulate AKT phosphorylation in mouse fibroblast [56]. Nevertheless, more studies, perhaps conducted in differentiated skeletal muscle and adipose tissue, could provide a better elucidation on the effect that the analogues have on the activation of the other key components of the insulin signalling pathway such as the downstream effectors like GLUT4. Furthermore, an in vivo study within the same report suggested that this analogue could still afford glycaemic control as per observation in the oral glucose tolerance test in mice. The analogue had a similar glucose-lowering effect as native insulin [56]. To further understand the glycaemic control profile of the analogue, long-term studies that map the effect that the analogue has on each component of the insulin signalling pathway and show the side effect profile of the analogue are pivotal. Not only would the studies validate the efficacy of the analogues but also uncover any potential toxicological challenges.

#### 4.2.3. Single-Chain Insulin

Despite being recognised as a precursor, proinsulin is technically a single-chain molecule. The hormone belongs to an ancestral metazoan superfamily of single-chain homologues, including vertebrate insulin-like growth factors (IGFs) and invertebrate neuroendocrine factors [57,58,59]. Therefore, understanding these evolutionary connections could be pivotal and hold a premise in contemplating and designing single-chain insulin (SCI) analogues. Suggested strategies include the replacement of the C-peptide molecule within the proinsulin with peptide or chemical linkers to merge the A and B chains by the N-terminal, thus synthesising an SCI analogue [60,61]. Earlier on, a functional application of the SCI was developed using apidic acid in place of C-peptide in proinsulin [58,62,63,64]. The synthesis of the SCI was linked directly by the peptide link, producing a seemingly identical structure to native insulin, but it was found to possess little to no blood glucose-lowering capability [65]. Earlier work on SCI laid a foundation for exploring more SCI analogues, as we have seen in the last decade.

Further investigation of potential chemical linkers of chain A to chain B ceased until the discovery of polyethene glycol (PEG) in 2013 (Figure 8). The PEG_12_, as a cross-linker, exhibited poor biological activity as an insulin molecule when the entirety of both chains was utilised. Retaining the same PEG molecule and the whole A chain resulted in an improvement in insulin activity that permitted the correct formation of interchain bonds [66]. Mao and colleagues have reported on an ultra-stable single-chain insulin analogue that resists thermal inactivation and exhibits a duration of biological signalling equivalent to the native protein [67]. In this study, the strategy aimed to directly link the A and B chains by introducing a short C domain (EEGPRR), thus forming 57 amino acid residues. The SCIs retained the binding efficiency to the activated insulin receptor. Interestingly, the SCIs demonstrated an attenuated ability to activate mitogenic 1. In vivo observations suggested that SCIs presented with pharmacodynamic properties similar to the short-acting insulin analogue Lispro. Most importantly, SCIs remained stable for 140 days and 48 h when incubated at 45 and 75 degrees centigrade, respectively [67].

The emergence of thermostable insulin represents a breakthrough that could circumvent the cold supply chain of insulin and other biologics. The need for refrigeration in part could threaten the therapeutic efficacy of insulin, especially for patients in areas that contend with an intermittent power supply. In addition, thermostable insulin could further lessen the burden of cool insulin storage conditions while travelling. Moreover, climate change associated with the increase in temperatures threatens its stability and therefore necessitates cold storage. Therefore, such attempts should be encouraged and invested in and warrants further developments in studies on long-term stability, the efficacy of glycaemic profiling, and toxicological profiles. 

### 4.3. Glucose-Responsive Insulins

Insulin-induced hypoglycaemia is the most serious acute complication associated with insulin therapy [68,69,70]. Once injected, insulin absorption into the bloodstream continues despite blood glucose concentration [71,72]. Therefore, this necessitates innovations towards developing insulins whose bioactivity is modulated by blood glucose concentration, matching insulin profiles with glycaemia. Several attempts to develop a glucose-responsive insulin (GRI) have been developed using glucose-triggering signals from lectins, glucose oxidases, glucose transporters, and phenylboronic acid (PBA).

#### 4.3.1. Saccharide Linked Insulins

Lectins such as mannose-binding receptors are known to bind the carbohydrate domain of glycoproteins [73,74,75]. Kaarsholm et al. sought to utilise this system to engineer an insulin analogue whose bioactivity is modulated by changes in blood glucose concentrations [76]. The premise behind the design was aimed at coupling insulin with a monosaccharide and then envisaging if such a complex would be recognised by the mannose receptor (MR). Affinity for the insulin–saccharide complex would lead to its internalisation and degradation by the lysosome. Since glucose itself can bind directly to the mannose receptor, it is theorised that high glucose concentration should antagonise recognition of this complex by the MR, leading to insulin availability, while a lower glucose concentration would lead to more clearance of the insulin complex. In this way, plasma insulin would therefore be modulated by glucose concentration, thus a glucose-responsive insulin. This study used human insulin with various saccharides such as mannose and fructose at A^1^, B^1^, and B^29^ [76]. The assembly of the saccharide–insulin complex has been achieved through the synthesis of insulin MK-2640, illustrated in Figure 9. MK-2640 on testing showed insulin receptor binding properties.

Competing binding assays on MR demonstrated that higher glucose concentration antagonises the saccharide–insulin complex, thus attenuating its degradation at high glucose concentrations. MK-2640 exhibited moderate to high systemic clearance in dogs and male Yucatan minipigs, 8 and 39 mL/min/kg, respectively, whilst the bioavailability varied between 26% and 44%. This study highlights the potential of saccharide-appended insulin analogues as a novel strategy for achieving glucose-responsive insulin therapy, even though more research is necessary to optimise the pharmacokinetic properties and improve the glucose responsiveness of these analogues. Though promising, the collation of long-term studies is necessary to further understand its ability to control blood glucose within an acceptable range.

#### 4.3.2. Phenylboronic Acid Insulin

By taking a page from the development of glargine related to altering insulin’s isoelectric point, this modification can change its solubility behaviour at various pH values. Qui et al. and Lin et al. have reported phenylboronic acid (PBA) as a glucose sensor, which can lead to the release of bioactive insulin only at higher blood glucose concentrations (Figure 10). This innovation was inspired by the ability of PBA to bind reversibly to cis-1,2 or cis-1,3 diols such as glucose, thus creating a negative charge on the boronic acid upon attachment to glucose [77,78]. The authors envisaged that adding PBA, particularly to Glargine, would lower the insulin isoelectric point and increase insulin solubility only when high interstitial glucose levels exist. Therefore, bioactive insulin would only be released in response to the hydrolysis or oxidation of functional groups found in PBA.

As illustrated by the phosphorylation of intracellular Akt, the authors confirm the PBA–insulins’ ability to initiate the insulin signalling cascade in a manner comparable to Glargine, confirming no activity loss. More activity of PBA–insulin was observed in a hyperglycaemic state than in a hypoglycaemic state, which further supported that PBA insulin was a glucose-response insulin. Crucially, PBA–insulin had higher activity under hyperglycaemic than under hypoglycaemic settings, highlighting its glucose-responsive characteristics. Considering these developments, long-term studies of PBA insulins are needed to evaluate their efficacy in improving glycaemic control, particularly their effect on glycated haemoglobin (Hba1C). Toxicity studies of boron have yielded no significant changes in the cardiovascular, respiratory, and gastrointestinal systems, suggesting a low toxicity profile [79]. Nevertheless, the known toxicity profile of boron does not negate the necessity for further toxicity studies.

### 4.4. Hepato-Preferential Insulin Analogues

Physiologically, the liver receives about three times more insulin than the peripheral tissues due to the first metabolic hepatic pass [2,80,81]. Subcutaneous insulin administration bypasses the liver, rendering it impossible to recreate the normal portal-to-arterial insulin gradient available due to a healthy pancreas [2,82,83]. As a result, this leaves the patient with arterial hyperinsulinemia and/or hepatic insulin deficiency. The latter deficiency could be associated with excessive glucose production, contributing to hyperglycaemia and eventually poor glycaemic control in some individuals [6]. Researchers have proposed developing a hepato-preferential insulin analogue that can restore proper distribution to fix this imbalance between hepatic and systemic insulin concentrations.

As depicted in Figure 11, insulin-327 is an insulin analogue of human insulin, modified by deleting threonine at position B30, substituting lysine at position B29 with arginine, adding lysine at position A22, and then attaching a 22-carbon length fatty diacid to A22; it was reported for the first time in 2012 by Brand and colleagues [84]. This design allows the insulin to reversibly bind strongly to plasma albumin. The unique feature of insulin-327 is its hepato-preferentiality action. Mechanistically, this is achieved because the analogue cannot pass through the tightly packed endothelial capillary walls in skeletal muscle tissue, whereas it can quickly enter the liver due to sinusoids. As a result, higher levels of insulin-327 accumulate in the liver compared to muscle and fat [2,84].

A further study on dogs compared how the same dose of insulin-327 through either the hepatic portal vein or a peripheral vein affected the liver’s ability to suppress glucose production and increase glucose uptake under hyperglycaemic postprandial conditions [2]. The response to insulin-327 infusion was more similar to that of insulin delivered through the portal vein than the peripheral vein. Insulin-327 successfully restored the normal glucose uptake by the liver. This is important because people with diabetes often have abnormal hepatic glucose uptake. The insulin analogue also corrected the balance of glucose uptake between the liver and muscle, restoring the normal 50/50 distribution. The hepato-preferential effects of 327 gradually decreased as the experiment progressed [2]. The strategy could improve diabetes management by restoring close physiological balance. The discovery of insulin 327 and its hepato-preferential effect is a testament to innovative strategies which aim to prioritise the management of diabetes within physiological limits.

More studies should consider understanding the effect of insulin-327 on the liver function and mitogenicity. This is because elevated insulin levels in the portal circulation enhance growth hormone receptor expression and signalling, which stimulates hepatic production of insulin-like growth factor-1 (IGF-I). These higher levels of IGF-I have been linked to a higher risk of various cancer types such as breast, colorectal, premenopausal breast cancer, and prostate cancer. The pharmacokinetic and pharmacodynamic studies should also be prioritised to ascertain its duration of action. Its metabolism and clearance studies should also be prioritised. Whether insulin-327 still undergoes similar renal elimination is subject to investigation considering its size, given the smaller fenestrations found in the glomerulus. From the application point of view, insulin-327 appears to be more appealing for type 1 diabetes, considering its lack of insulin. However, type 2 diabetes requiring insulin therapy could equally benefit. The effort complements approaches that aim to ensure the organ’s physiological needs are met. With the discovery of oral insulin formulations which could allow insulin to enter the portal vein, mimicking a physiological process, insulin-327 represents a strategy in which the hepatic needs for insulin could be met.

### 4.5. Once-Weekly Insulin Analogues on the Verge of Approval

Researchers have leveraged some of the insights from basic peptide design strategies, culminating in another two insulin analogues. A few once-weekly insulin therapies are currently in advanced clinical stages where the inspiration for the design has been drawn out from the likes of Glargine, detemir, degludec, insulin-327, insulin-406, and Peglispro. The proposed analogues include insulin Icodec and Efsitora alfa developed by Novordisk and Eli Lilly, respectively [85,86]. However, these analogues have yet to gain regulatory approval from the regulatory bodies, despite extensive clinical developments.

The three-dimensional molecular design of Icodec insulin analogue resembles that of detemir and degludec because much like detemir and degludec, it is acylated with a carbon chain. As seen in Figure 12, Icodec is acylated with icosane, a C20 fatty acid at B^29^ [87]. The acetylation extends the time-action controlled release in the injection site depot through the formation of hexamers and a strong complexation with human serum albumin (HSA), forming sizeable hydrodynamic-size particles [87]. These attributes thus ensure a decreased availability of insulin monomer, which is biologically active. The Icodec comes with the following amino acid substitutions TryB^16^ and PheB^25^ for His and Tyr^14^ for Glu, which have been shown to reduce the IR affinity whilst improving the stability [88]. Moreover, in vitro studies suggest that it lacks mitogenic activity, which is important considering that insulin analogues could elicit a mitogenetic risk [87,88]. Pharmacokinetically, Icodec exhibits a T_max_ of 18 h and a boost with an extended half-life of 196 h [89,90]. The Icodec insulin analogue has undergone all three clinical phases, where Glargine was used as the comparator to understand the efficacy and safety profiles of Icodec [91]. Emerging data suggests that it has a slow onset of actions, with the action noticeable from day 3 post once-weekly injection [92]. Nonetheless, compared to Glargine, Icodec is slightly superior, judging by its efficacy in attaining lower HbA1c and fasting blood glucose at the end of the study. Although hypoglycaemia was observed with Icodec, statistical analysis suggested insignificant incidences, which were not severe [86,93,94].

The three-dimensional molecular design Efsitora insulin analogue is the epitome of peptide design and engineering (Figure 13). Efsitora’s design is inspired by background knowledge of single-chain insulins (SCIs) and their extended stability. Structurally, Efsitora is attained through covalently binding both A and B chains to create SCI. It also comes with nine amino acid substitutions: TyrB^16^ → Glu, IleA^10^ → Thr, TyrA^14^ → Asp, AspA^2^ → Gly, ProB^28^ → Gly, LysBLysB^29^ → Gly, TherB^30^ → Gly, IleA^10^ → Thr, and AspA^2^ → Gly. The C terminus of the peptide is connected to an IgG2 Fc domain [85]. The fusion with the Fc domain contributes to the extended half through recycling. Elaboratively, the Fc domain appended allows insulin Efsitora to avoid pinocytosis and subsequent degradation by the endosomes [91]. Similar to Icodec, Efsitora produces significant hydrodynamic substances. The design of Efsitora, therefore, aims to create a systemic reservoir for insulin.

Efsitora has been shown to reach T_max_ after 96 h and a remarkable half-life of approximately 17 days [91]. Clinical studies have demonstrated the efficacy of Efsitora in attaining desired glycaemic outcomes. In these studies, once daily, Degludec or Glargine injection was used as a comparator, and the efficacy of Efsitora was comparable for interventions on glycaemic outcomes. Additionally, hypoglycaemic episodes were observed to be 25% lower compared to Degludec insulin analogue [96,97,98]. The key outcomes from the clinical trials are presented in Table 2 and Table 3.

These ultra-long-acting insulins can significantly reduce the injection times from 365 to 52 times a year. This could mark a massive shift in the management of diabetes, considering the alleviation of the psychological burden that comes with frequent injections. The reduction in injection time, could potentially see a reduction in insulin costs, making it universally accessible. Most importantly, these efforts could militate patient non-compliance, which could be paramount in the onset and progression delay of diabetes associated complications such as cardiovascular diseases, amputations, nephropathy, neuropathy, and retinopathy. Uncontrolled hyperglycaemia is central to the development of these complications, as such, Icodec and Efsitora could be instrumental in the prevention of these complications, resulting in reduced mortality and morbidity associated with diabetes. Perhaps once approved, beyond studying glycaemic control, the onset and progression delay of complications mentioned above should also be studied. In view of the challenges associated with the discovery and development of non-invasive insulin delivery formulations, a weekly insulin injection should be considered. However, the pursuit of non-invasive insulin formulations should still be sought.

## 5. Author Perspective and Recommendations

The discovery of insulin is a monumental discovery in the management of diabetes. The heightened understanding of diabetes mellitus coupled with innovations in insulin design and engineering have demonstrated the possibility of continuous improvement in insulin therapy, which have yielded better diabetes management. The design and production technology of insulin with varied pharmacokinetic behaviour has revolutionised diabetes mellitus management. The research and innovation in insulin therapy are fast-paced, with insulin with extended pharmacokinetics, ultra-stable insulin, glucose-responsive insulin, and hepato-preferential insulin in the pipeline. These developments have been made possible in part through a greater understanding of the three-dimensional molecular structure of insulin as well as its behaviour. Improving insulin pharmacokinetic stability has the potential to decrease insulin injections whilst affording adequate glycaemic control.

Insulin is a thermolabile protein, whose degradation increases with increase in storage temperature. Studies have reported that insulin should be stored in the refrigerator at 2–8 °C [99]. These storage requirements pose a difficult challenge to families in low-income areas with hot climate and lack access to refrigeration as reported by the IDF in the Life for a Child Index study [100]. This underscores the need for an ultra-stable insulin analogue of which stability does not rely on refrigeration. The proposed thermostable analogues reported in this article will benefit more from studies that can show their stability profiles under conditions that simulate torrid zones. This will adequately inform our understanding of how these conditions might affect the potency of these analogues as well as the propensity to form fibrils. The development of an analogue capable of retaining efficacy in high temperatures will be a game changer not only in areas such as Sudan, Ethiopia, and Haiti, where they use clay pots and goat skins as storage and cooling devices for insulin [101], but also in many other low- and middle-income countries facing similar challenges. Only 5.3% of low-income countries and 33.3% of lower-middle income countries report that more than two-thirds of families have access to a home refrigerator, which is essential for maintaining insulin potency [98]. This means that approximately 95% of low-income and two-thirds of lower-middle income countries would benefit substantially from thermostable insulin analogues that do not require refrigeration. The long-term cost savings that will stem from thermostable insulin analogues include reduced rates of diabetes-related complications that arise from the loss of insulin of potency due to suboptimal storage conditions.

The currently used pharmacological agents in the management of diabetes includes oral antidiabetics and insulin injections [102]. However, insulin’s use is associated with an increased risk of hypoglycaemia, especially with the short-acting analogues. The fear of hypoglycaemia is one of the reported reasons that leads to non-adherence to insulin therapy [103]. Despite the benefit of intensive glycaemic control in reducing the risk of microvascular complications, it has been shown to cause a threefold increased risk of severe hypoglycaemia. Although considerable preventive strategies such as patient education, glucose monitoring, dietary adjustments, and close medical supervision have been proposed and implemented, hypoglycaemia is still a challenge [104,105,106]. In fact, yearly rates of severe hypoglycaemia in patients with type 1 diabetes have been reported to range between 3% and 13%. Given these challenges, glucose-responsive insulin analogues present a promising strategy for addressing hypoglycaemia. By allowing insulin release to correspond more closely with blood glucose levels, these analogues can help minimise the frequency and severity of hypoglycaemic episodes. Lastly, the emergence of hepato-preferential insulin is a promising strategy that could mimic endogenous insulin secretion and circulation. Clinicians have primarily overlooked how insulin’s natural secretion from the pancreas into the portal vein has a concomitant effect on the liver. From the approaches above, perhaps the goal should be to leverage on the unique strengths presented by each analogue to develop insulins that can exhibit the above-mentioned characteristics. By carefully analysing the key advantages inherent in their design and engineering strategies, scientists and clinicians can collaborate in investigating the possibility of developing more effective insulin analogues. Coupling innovative insulin design, formulation, and delivery strategies could yield analogues with improved pharmacokinetics, stability, and patient adherence. With a heightened understanding of glucose metabolism, insulin pharmacokinetics, insulin structure and behaviour, and peptide design and synthesis technology, our goal should be to strive to develop a sustainable and patient-friendly insulin therapy which can provide a more permanent solution to diabetes mellitus. Diabetes is a chronic condition meaning that those on insulin therapy must inject themselves daily. The pain associated with injection has been cited alongside hypoglycaemia as a factor contributing to non-adherence. In this context, the development of once-weekly insulins, such as Icodec and Efsitora, provides a solution to this treatment burden. By reducing the frequency of injections from 365 to just 52 per year, these formulations could potentially improve both adherence and quality of life for diabetic patients. Considering the chronic nature of the disease, careful attention should focus on potential toxicity and safety profiles. Epidemiological data suggest that diabetes mellitus continues to face an upward trajectory despite accelerated preventative strategies. Therefore, efforts to improve insulin therapy should still be prioritised. Although the highlighted analogues above present unique and promising strategies in novel insulins development, potential challenges and limitations still exist. Synthesis of some of the insulin analogues reviewed above is associated with a complicated synthesis procedure, i.e., FMOC-based proteins synthesis and chemical oxidation techniques which could be associated with high cost, thus from a mass production point of view, there are scalability challenges especially when the goal is to produce affordable insulins. High cost of long-acting insulin analogues has already discouraged their widespread use in public health systems of countries such as Brazil, Bangladesh, South Africa, and Zambia [101]. For instance, in Brazil, the use of these analogues remains limited because their benefits were deemed insufficient to justify their price [102]. In light of these foreseen limitations, attempts should be made to explore the incorporation of recombinant DNA technology in the synthesis of novel insulin analogues. Whilst pursuing the innovative design and development, efforts should also be directed to exploring alternative delivery systems. Insulins as injectables in part hinders insulin therapy. Therefore, while the novelty in design is lauded, it should be coupled with novel formulating strategies that could see oral insulin administration being possible, for an example. It is therefore envisaged that the collaboration between peptide design and formulation scientists could be instrumental in achieving this goal.

## 6. Conclusions

The heightened understanding of insulin structure and behaviour continues to yield significant breakthroughs in the design of novel insulin analogues with varied pharmacokinetic and pharmacodynamic profiles. This has significantly improved the management of diabetes mellitus. The innovations in insulin design continue to demonstrate reliance and intent in revolutionising diabetes mellitus management beyond the desirable emergence of novel insulin analogues, presenting promising advancements in addressing the current side effects and challenges associated with insulin therapy. This paper highlighted recent developments in insulin analogue design and effectiveness, which holds the potential to unlock new treatment possibilities and contribute to the ongoing efforts in diabetes management. These emerging insulin analogues range from ultra-acting insulin, thermostable glucose-response insulins to hepato-preferential insulins as well as the once-weekly insulin Icodec and Efsitora, which are at the advanced clinical development stages. The design of this insulin continues to demonstrate scientific and clinical intent in improving diabetes mellitus management.

Significantly, more studies must be conducted in order to accelerate the approval of these analogues to the market. Substantial evidence has been brought forth regarding the relevance and necessity of these analogues as they have the potential to revolutionise glycaemic control as it is known. However, critical studies elucidating pharmacokinetic parameters of each analogue, particularly, the volume of distribution, renal and hepatic plasma clearance, and rate and extent of distribution. Further clinical studies should be conducted to ascertain glycaemic control compared to conventional analogue and conduct drug therapeutic monitoring. In addition, toxicity studies, including risk of cancers should be included and thereafter, dosages can be determined. Lastly, the crystal and conformation studies should be prioritised and envisaged to provide heightened understanding of the behaviour of novel insulin analogues. Such parameters would aid the advancement of these novel insulin analogues from concept to reality, changing diabetic treatment as we know it.

## Figures and Tables

**Figure 1 pharmaceutics-17-01239-f001:**
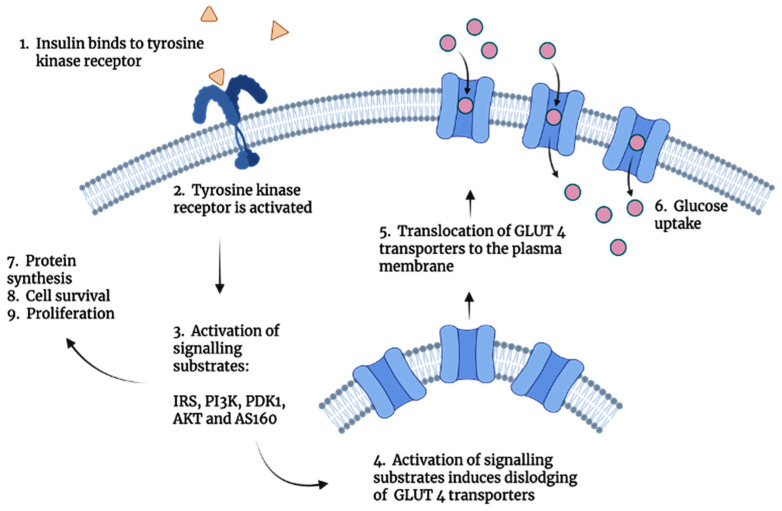
Schematic illustration of the insulin signalling transduction pathway. The initial steps within the pathway speak to the activation of signalling substrates, which facilitate the translocation of GLUT 4 transporters in the final steps.

**Figure 2 pharmaceutics-17-01239-f002:**
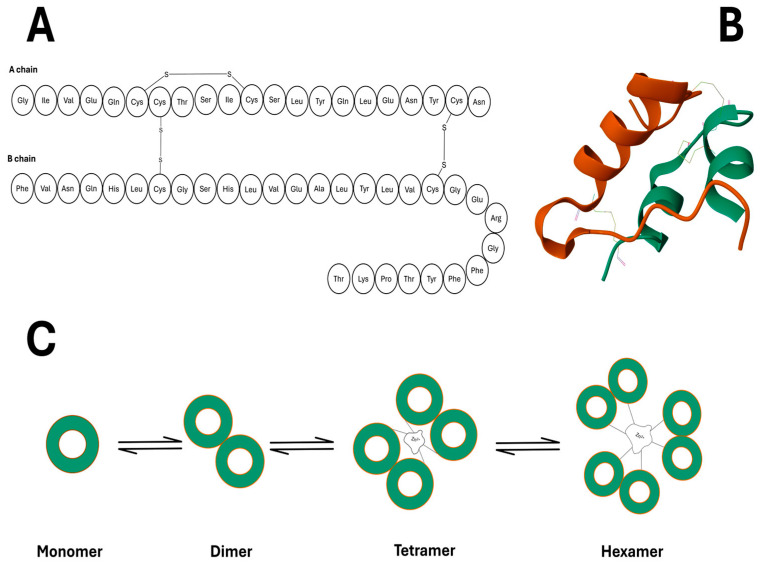
(**A**) Amino acid sequence of insulin monomers, consisting of A and B chains. (**B**) Three-dimensional representation of insulin, highlighting helical regions and disulphide bonds PDB code 3I40. (**C**) Stepwise aggregation of insulin monomers into dimers and subsequent tetramer and hexamer formation.

**Figure 3 pharmaceutics-17-01239-f003:**
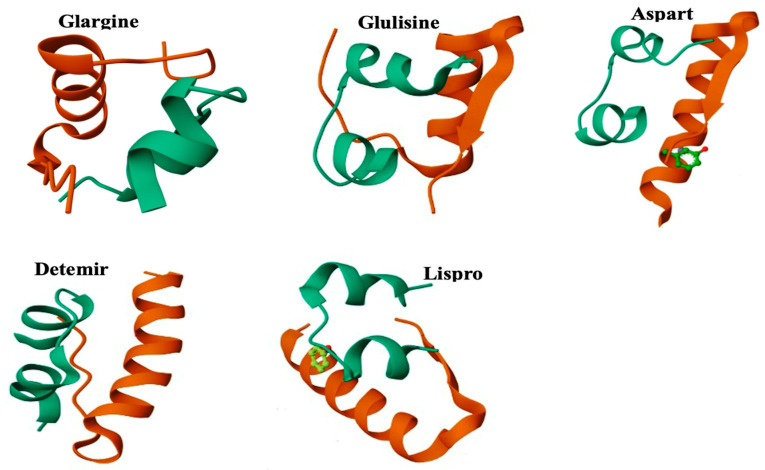
Crystallographic diagrams and illustration of amino sequence of conventional insulin analogues, Glargine, Glulisine, Aspart, Detemir, and Lispro. Protein database 5vis, 6gv0, 4gbl,9civ, and 5udp.

**Figure 4 pharmaceutics-17-01239-f004:**
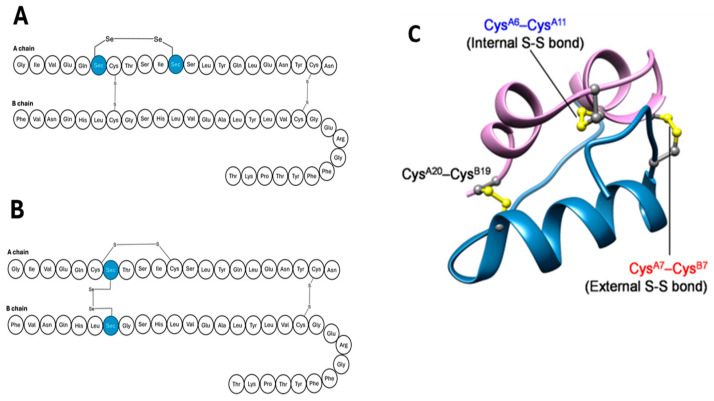
Amino acid sequence of two seleno-insulins having an external Se-Se bond (**A**) or internal Se-Se bond (**B**). (**C**) depicts a 3-D structure of seleno-insulin, where internal and external selenide bonds permit insulin folding [47].

**Figure 5 pharmaceutics-17-01239-f005:**
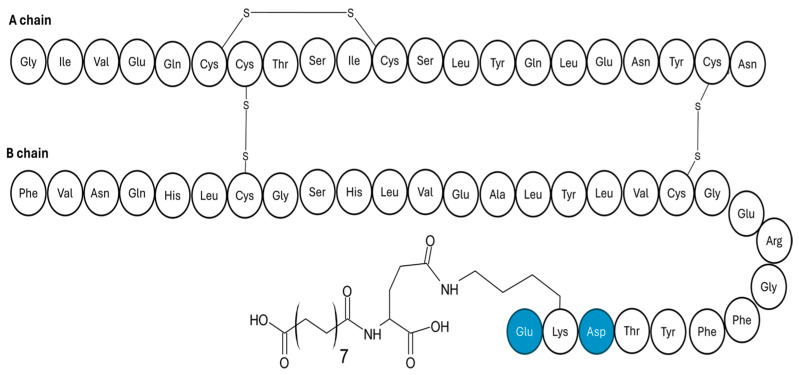
Amino acid sequence of INS601, showing the hexadecenoic acid appended at LYS29, and the amino acid substitutions (in blue), ProB28Asp and ThrB30Glu.

**Figure 6 pharmaceutics-17-01239-f006:**
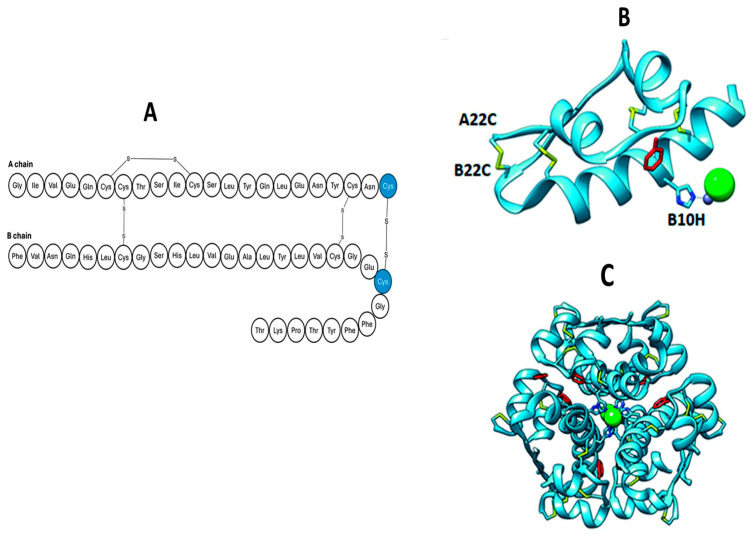
Amino acid sequence for the insulin analogue with an additional disulphide bond at A^22^–B^22^ (**A**). (**B**,**C**) demonstrate the crystal structure of the monomer and hexameric unit, respectively.

**Figure 7 pharmaceutics-17-01239-f007:**
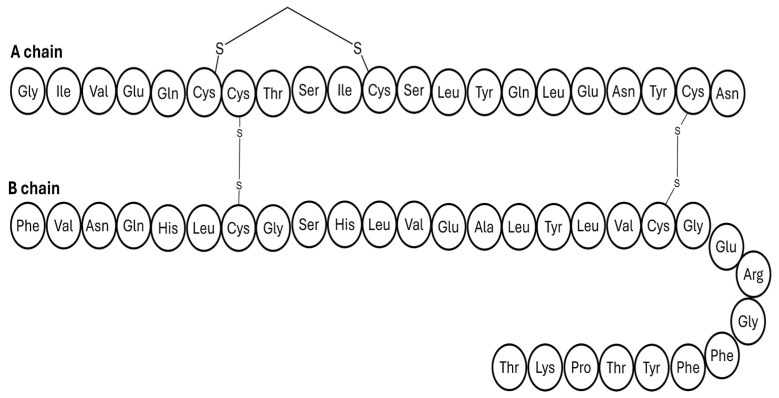
Amino acid sequence of a methylene thioacetal human insulin analogue where the A^6^–A^11^ disulphide bond is replaced with the non-reducible methylene thioacetal to stabilise the N-terminus of the A chain.

**Figure 8 pharmaceutics-17-01239-f008:**
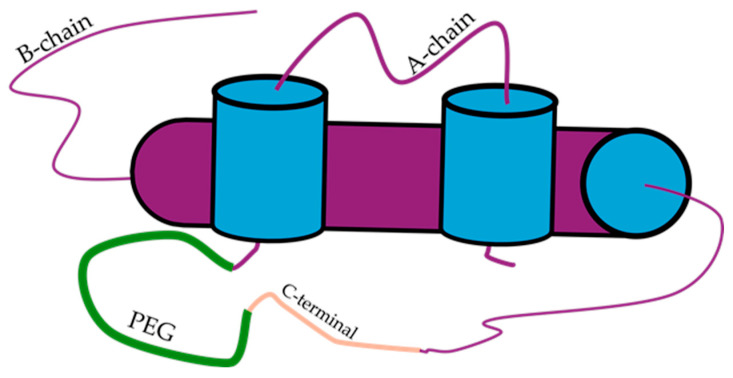
A visual representation of pegylated insulin where C domain has been incorporated [66].

**Figure 9 pharmaceutics-17-01239-f009:**
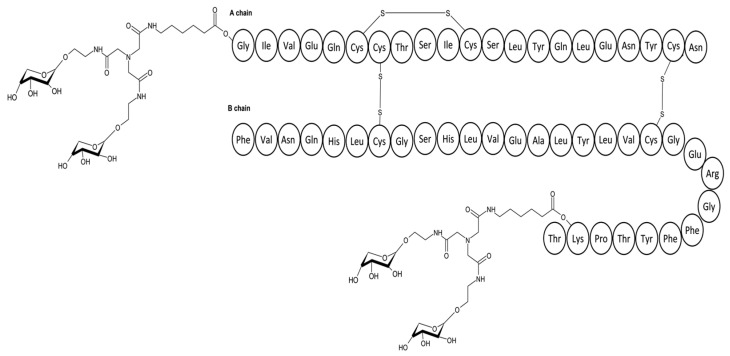
Amino acid sequence of the MK-2640 analogue achieved through coupling human insulin with monosaccharides.

**Figure 10 pharmaceutics-17-01239-f010:**
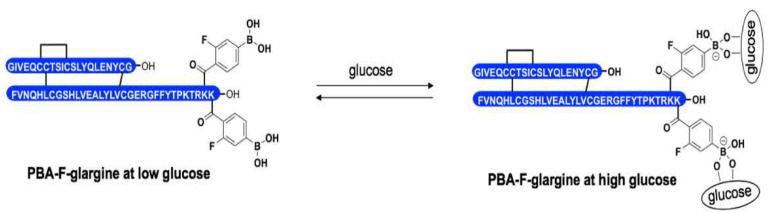
A representation of insulin glargine coupled with PBA, where binding of PBA can be reversed by high glucose concentrations [77].

**Figure 11 pharmaceutics-17-01239-f011:**
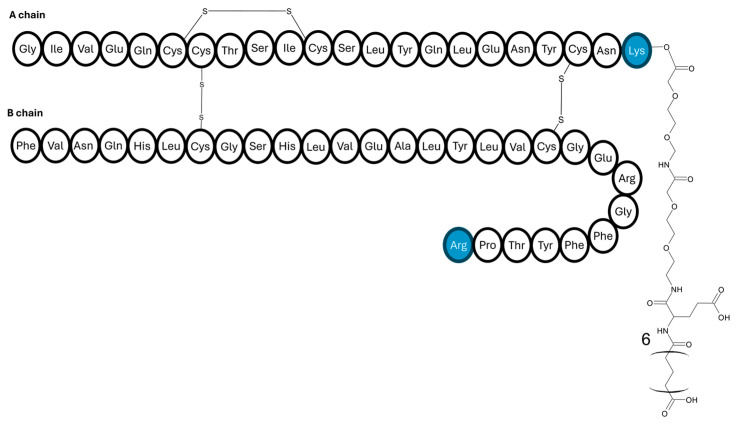
Amino acid sequence of the insulin-327 analogue. The human insulin is modified by deleting threonine at position B30, substituting lysine at position B29 with arginine, and adding lysine at position A22 and a 22-carbon length fatty diacid.

**Figure 12 pharmaceutics-17-01239-f012:**
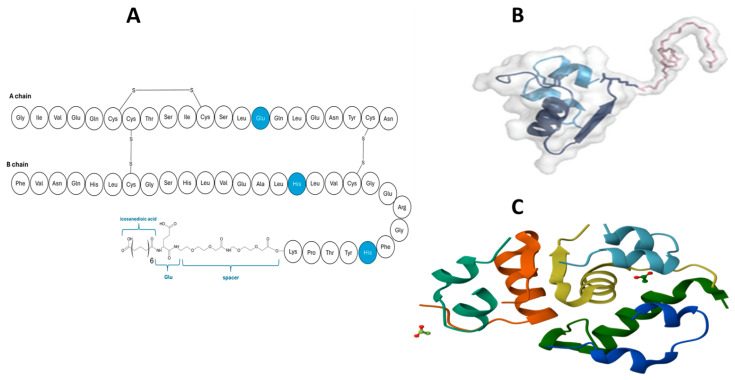
(**A**) presents the amino acid sequences of the two, once-weekly insulin Icodec. (**B**) is a representation of 3D structure, and (**C**) represents the crystallographic of insulin icodec hexameric unit [95].

**Figure 13 pharmaceutics-17-01239-f013:**
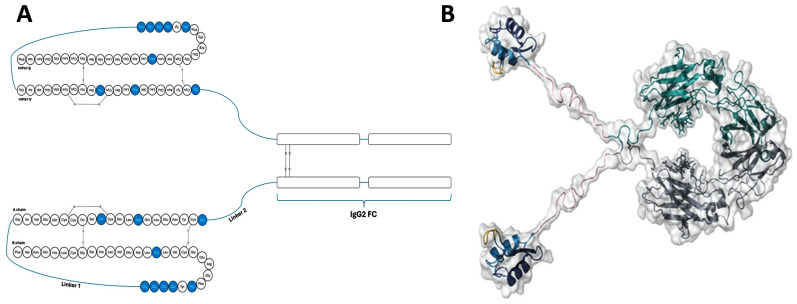
A representation of amino acid sequence of insulin molecules attached to the IgG2F (**A**) and 3D Efsitora 3D structure depicting conformations (**B**) [95].

**Table 1 pharmaceutics-17-01239-t001:** Summary of the insulin analogues design and effects.

Insulin Analogue	Design	Pharmacokinetic and Pharmacodynamic Effects
**Seleno-insulin**	Replacing the solvent-exposed disulphide bridge with the diselenide bridge at CysA^7^-CysB^8^.	Improved kinetic and thermostability. Resistance to degradation by IDE. Ease in aggregation of seleno-substituted insulin monomers. Prolongation of half-life from 1 to 8 h.
	Substitution of intrachain disulphide bridge to form diselenide CysA^6^-CysA^11^.	Heightened the thermodynamic and kinetic stability of insulin monomer. Minimised hydrolysis by IDE.
**Degludec-like insulins (INS061)**	Acylation of LysB^29^ with hexadecenoic acid and substituting AspB^28^ and GluB^30^ with ProB^28^ and ThrB^30^.	Increase in oligomerisation into hexamers. Sustained blood glucose-lowering effect for 25 h duration in streptozotocin-induced diabetic rats. Quick absorption and slower elimination after subcutaneous administration.
**Additional disulphide bonds**	Additional disulphide bond (4th) on an extended A chain (24 amino acids) and truncated B chain (22 amino acids) (A^22^–B^22^).	Results from a stress test indicated added stability, as aggregates could only form after 42 h compared to 10 h achieved by native insulin.
**Carbon-based Linkers**	Replacing the intrachain (A^6^–A^11^) disulphide bond with a methylene thioacetal bond.	Improved thermostability over native insulin when incubated at 60 °C for 4 days. Results from a stress test indicated added stability, as no aggregation occurred before reaching a 100 h incubation period. Blood glucose-lowering capabilities were comparable to those of native insulin.
**Single-chain insulin (SCI)**	Replacing the C-peptide in proinsulin with apidic acid and combining the two chains at A^1^ and B^29^.	Less potent than native insulin. Increased thermal stability. Decreased structural flexibility.
	Linking A and B chains with a shorter than usual C domain (EEGPRR) form a 57 amino acid chain.	Marked resistance to thermal inactivation in vitro that is compatible with native duration of activity in vivo Limited SCI aggregation in the concentration range 1–7 mM.
**Saccharide linked insulin**	Coupling insulin with mannose or fructose at A^1^, B^1^ and B^29^.	Saccharide–insulin complex showed insulin receptor binding with a lower affinity than native insulin. The complex could bind to the mannose receptor. The complex could not precipitate an inflammatory response.
**Phenylboronic acid (PBA)-insulin**	Incorporation of phenylboronic acid on Glargine to act as a glucose sensor.	Glucose control is superior to native insulin. PBA–insulin showed higher activity under hyperglycaemic than hypoglycaemic states, indicating glucose-responsive characteristics.
**Hepato-preferential insulin analogue**	Attaching a 22-carbon length fatty diacid.	Hepato-preferentiality. Restored the normal glucose uptake by the liver. Lowered levels of arterial glycerol and free fatty acids in the blood.
**Insulin Icodec**	Acylated by 20 carbon fatty acid.	Injected once weekly. Attenuated IR binding. Attains desired glycaemic control.
**Insulin Efsitora**	Fc bound.	Injected once weekly. Attenuated IR binding. Attains desired glycaemic control.

**Table 2 pharmaceutics-17-01239-t002:** Summary of outcomes from ONWARDS 1–6 clinical trials for insulin analogue Icodec. S represents number of severe hypoglycaemic episodes and Ćś is the number of clinically significant hypoglycemic episodes.

Trial Acronym, (NCT Number)	Duration (Weeks)	Background DM and Therapy	n	Treatment Group	% Change in HbA1c	Number of Hypoglycemic Episodes		Sponsor
						Š	Ćś	
ONWARDS1 (NCT04460885)	78	Type 2 DM, insulin naive	984	Insulin Icodec Insulin Glargine	−1.55 −1.35	1 3	143 75	Novo Nordisk A/S, Bagsværd, Denmark
ONWARDS 2 (NCT04770532)	26	Type 2DM, insulin switch	526	Insulin Icodec Insulin Degludec	−0.93 −0.71	0 1	113 41	Novo Nordisk A/S
ONWARDS 3 (NCT04795531)	26	Type 2DM, insulin naive	588	Insulin Icodec Insulin Degludec	−1.57 −1.36	0 2	53 23	Novo Nordisk A/S
ONWARDS 4 (NCT04880850)	26	Type 2DM, basal-bolus therapy	582	Insulin Icodec + Insulin Aspart Insulin Glargine + Insulin Aspart	−1.16 −1.18	7 3	937 935	Novo Nordisk A/S
ONWARDS 5 (NCT04760626)	52	Type 2 insulin naïve	1085	Insulin icodec Insulin glargine or insulin degludec	−1.68 −1.31	0 5	104 76	Novo Nordisk A/S
ONWARDS 6 (NCT04848480)	52	T1DM, basal-bolus therapy	582	Insulin icodec + insulin Aspart Insulin degludec + insulin Aspart	−0.37 −0.54	56 25	5047 2811	Novo Nordisk A/S

**Table 3 pharmaceutics-17-01239-t003:** Summary of outcomes from QWINT 1–5 clinical trials for insulin analogue Efsitora.

Trial Acronym, (NCT Number)	Duration (Weeks)	Background DM and Therapy	n	Treatment Group	% Change in HbA1c	Hypoglycemic Event Rate (%)	Sponsor
QWINT 1 (NCT05662332)	52	T2DM, insulin naive	715	500 U/mL Insulin Efsitora Alfa 100 U/mL Insulin Glargine	−1.19 −1.16	0.50 0.88	Eli Lilly and Company, Indianapolis, IN, USA
QWINT 2 (NCT05362058)	52	T2DM, insulin naive	921	500 U/mL Insulin Efsitora Alfa 100 U/mL Insulin Degludec	−1.26 −1.17	0.58 0.45	Eli Lilly and Company
QWINT 3 (NCT05275400)	26	T2DM, insulin switch.	977	500 U/mL Insulin Efsitora Alfa 100 U/mL Insulin Degludec	−0.81 −0.72	0.84 0.74	Eli Lilly and Company
QWINT 4 (NCT05462756)	26	T2DM, basal-bolus therapy	723	500 U/mL Insulin Efsitora Alfa + 100 U/mL insulin lispro 100 U/mL Insulin Glargine + 100 U/mL insulin lispro	−1.01 −1.00	6.58 5.94	Eli Lilly and Company
QWINT 5 (NCT05463744)	26	T1DM, basal-bolus therapy	689	500 U/mL Insulin Efsitora Alfa 100 U/mL Insulin Degludec	−0.51 −0.56	14.03 11.59	Eli Lilly and Company

## Data Availability

No new data was generated.

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
