# Peer review of "Emerging Insulin Analogues: A Glimpse into How Insulin Analogues May Look in the near Future"

_pharmaceutics, 2025, doi:10.3390/pharmaceutics17101239_

Round 1

Reviewer 1 Report

Comments and Suggestions for Authors

I read with great interest the review entitled “Emerging insulin analogues: A glimpse into how insulin analogues may look in the near future” which made a clear descriptions of novel insulin engineering strategies in addition to the effective use of figures to simplify the complex molecular structures.

Overall, I find the review to be well-structured and informative, with major issues that require revision. Here is a summary of my review:

  1. This review does not state its search strategy, so I recommend adding a "Methods of Literature Search" section outlining literature search strategy, databases used (PubMed, Scopus, …..), keywords, and inclusion/exclusion criteria for study.
  2. The sequence of figure numbers seems to be incorrect. Figures 1, 2, 3, 4, 5, 6, and 8 are cited in the manuscript; however, Figure 7 is absent, and Figure 3 is mislabeled as Figure 2 in the text (page 7). Please carefully review and correct the text's citations and figure numbering.
  3. I advise the authors to include a more critical assessment for each class of emerging analogs in order to change the tone of the manuscript from a descriptive only to a scientific review. As the authors used repeated concluding sentences during the discussion of different analogues (e.g., long-term studies are needed) which used several times. For example, you can focus mainly on the impact hepato-preferential insulins on the liver function, mitogencity, or the impact of the high cost of some analogues on their accessibility and utility in clinical sittings an so on…….
  4. Line 302-304: “… more studies perhaps conducted in differentiated skeletal muscle and adipose tissue could provide a better elucidation on the effect the analogs on insulin signalling pathway activation. Further in vivo studies …” please mention these studies.
  5. The final section about the authors’ conclusion effectively summarizes the main findings and provides a clear and concise summary. Consider adding a paragraph on future research directions and the limitations rather than restating the need for long-term studies. This would help provide a sense of continuity and identify further areas for investigation.
  6. Lines 630-639 contain instructional text from a journal template. Please remove it.
  7. Some references are inconsistently formatted. For example, reference 62: "JOURNAL OF BIOLOGlCAL CHEMISTRY". Please ensure all references follow MDPI’s guidelines.

Reviewer 2 Report

Comments and Suggestions for Authors

The authors have written a review detailing the development and deployment of various insulin

analogues to treat diabetes mellitus (type 1 and late-stage type 2). These progressions are contextualised by the inherent thermolability/thermosensitivity of naturally occurring insulin and the place this holds in insulin resistance, as an increasingly common pathophysiological phenomenon, closely related to a variety of metabolic diseases including diabetes and obesity. The insulin analogues discussed in the review appear to have increased thermostability and an extended pharmacokinetic profile. While bioavailability and half-life are requisite measures of pharmacological action it would have been interesting to compare other parameters such as the elimination rate constant, the volume of distribution, plasma clearance (hepatic and renal) etc. as a validation of the adoption of these analogues in a viable treatment protocol. The game changer of reducing the subcutaneous administration of insulin analogues to once a week is particularly noteworthy, as these analogues can be designed to be less susceptible to aggregation and fibrillation, where in the naturally occurring form this can be at best detrimental to the surveillance of HbA1c as a marker for average glucose levels and in worse-case scenarios can have long-term harmful effects.

The authors describe the role that insulin plays in regulating many pathways at a molecular level. This narrative should be supported with a schematic representation as it would help the reader to understand why insulin analogues have the potential to militate against some of the aberrant mechanistic disturbances that can lead to the disease state. The rendition(s) of the three dimensional structure of insulin (referred to as the ‘architecture’ of the molecule) need to be improved. While the authors discuss the dynamics of the aggregation that lead to the resolution of the hexamer, there are quite detailed spatial resolutions available that also highlight predicted receptor-binding surfaces that can overlap insulin’s hexamer-forming surface. These also highlight the so-called “classical” set of receptor-binding residues that are not really shown in the diagrammatic representations used in the current manuscript. The diagrammatic representations of the insulin analogues featured in the review don’t convey the full story, particularly the way in which these molecules might engage with the insulin receptor in a remarkably complex way, as is the case with naturally occurring insulin. The authors discussion of INS061 is a case in point, where this analogue was described as ‘retaining receptor binding affinity’ with similar structural characteristics. Structural representations based solely on amino acid sequences (with interstitial modifications where appropriate) somewhat disconnect from the interesting folding mechanisms that are discussed in the narrative. For instance it would be fascinating to observe how the MK-2640 analogue with all of its potential conformational flexibility arranges three-dimensionally given some of its promising pharmacological performance in a pre-clinical model.

The authors describe in detail the advent of the insulin-327 analogue with its hepato-preferential effects (regarded as a seminal breakthrough). Ironically the authors chose not to show the structure of this molecule and therefore exemplify the attributes that make this a significant innovation. This part of the draft progresses to the discussion of insulin analogues (administered once-a-week) that are reported to be in late-stage clinical development. The authors could have taken the opportunity to summarise (in another table) the number of participants in the trial and under whose auspices these have been conducted. The three-dimensional structure of ‘Etsifora’ (and not its ‘architectural design’) is actually more nuanced than the rendition shown in Figure 8 (as is similarly the case with ‘Icodec’) and deserves to be shown in its dynamic form, given the report of its efficacy in lowering HbA1c, the widely accepted marker for average glucose levels. The insulin analogues encapsulated in ‘Table 1’ are creditable but the PK and PD effects that are described are not sufficiently granular and need to be articulated in comparison with appropriate controls. It wasn’t clear whether any of the ‘ultra-long-acting’ insulin analogues had gained regulatory approval, as there is no doubt that fewer subcutaneous administrations to achieve the same order of control would be desirable.

It is debateable what the statement, ‘The discovery of insulin remains the monumental discovery in the management of insulin’ actually means, however the authors are right to highlight the need for  the development of thermostable analogues for deployment in those areas that are challenged by the effects of climate change. One suspects that many more nation states face similar environmental challenges as the authors cite for Sudan, Ethiopia and Haiti. There is no doubt that the incidence of diabetes mellitus (in particular) is increasing at a worrying rate and therefore investment in ‘stable’ innovations to combat the proliferation of the disease must be prioritised. The authors haven’t included any modelling data to show what might happen if the regions most afflicted had access to insulin analogues, but the development of pharmaceutical payloads that facilitate ‘the oral administration’ of insulin analogues would be an extraordinary breakthrough.

I have also included a list of corrections as follows:

Line 18 – Should read, ‘…this review consolidates’…

Line 21 – Should read, ‘Although the analogues are often limited to preclinical studies…..’

Line 24 – Should read, ‘are glucose-responsive, and are hepato-preferential insulin analogues’.

Line 24 – Should read, ‘Due to the fast pace of innovation in the design of insulin analogues….’

Line 27 – Should read, ‘…they aim to strive for better……’

Line 54 – Should read, ‘….to the non-adherence of medication….’

Line 57 – Should read, ‘A chief side-effect of insulin therapy is hypoglycaemia’.

Line 68 – Should read, ‘…contributes to hyperglycaemia in patients…..’

Line 82 – Should read, ‘Proinsulin then exits the ER….’

Line 91 – Should read, ‘..released into the circulation….’

Line 92 – Should read, ‘….culminating in the translocation of glucose transporter…..’

Line 128 – Should read, ‘….it’s isoelectric character, solubility…..’

Line 139 – Should read, ‘Properties of Glulisine are all attributed…..’

Line 150 – Should read, ‘The modifications instituted for the detemir formulation….’

Line 151 – Should read, ‘Further development of…..’

Line 153 – Should read, ‘Degludec consists of hexadecanedioc acid…..’

Line 156 – Should read, ‘…half-lie of this insulin analogue,…..’

Line 170 – Should read, ‘The emergence of Degludec in a clinical setting….’

Line 176 – Should read, ‘……design of a seleno insulin analogue…..’

Line 188 – Should read, ‘…retention of insulin activity.’

Line 189 – Is this ‘Akt and GS3K in Hela cells at 1 mm concentration’?

Line 191 – Should read, ‘…abolishes the need for stabilisers…..’

Line 191 to Line 192  – Should read, ‘eases formulation stability during the manufacturing process.’

Line 197 – Should read, ‘….more studies focussed on the exploration of primary insulin sensitive cells….’

Line 207 – correct spelling whist should be whilst or while

Line 207 – Should read, ‘…to sustain a glucose lowering effect…..’

Line 209 – Should read, ‘..could be pivotal in facilitating the extended…..’

Line 212 – Should read, ‘…together with their potential toxicological effects…..’

Line 213 – Should read, ‘…for example cardiovascular, hepatic and renal hazards.’

Line 214 – Does not make sense, what are you trying to say here? Do you mean from an application point of view?

Line 228 – Should read, ‘…has since been named INS061.’

Line 233 – Should read, ‘…better AUC results than Degludec.’

Line 250 – Should read, ‘…can lead to a reduction in insulin availability……’

Line 251 – Should read, ‘….to self-assemble….’

Line 254 – Should read, ‘….in the development of the following categories……’

Line 258 – Should read, ‘One of the strategies been explored…..’

Line 268 – Should read, ‘…and the B chain truncated…’

Line 270 – Should read, ‘..between them being….’

Line 276 – Should read, ‘although this data was promising….’

Line 277 – Should read, ‘…elucidate the glycaemic control profile….’

Line 278 – Should read, ‘…using a stress ageing assay…..’

Line 280 – Should read, ‘…no reports on stability in relation to storage over time….’

Line 281 – Should read, ‘…should prioritise the understanding of the effect of long-term storage at various temperatures.’

Line 290 – Which researchers? The names need to be stated, as well as supported with a reference.

Line 298 – Should be a gap between the number and the unit, i.e. 60 oC.

Line 299 – Should read, ‘…by a stress-aging assay, in which no insulin…..’

Line 300 – Should read, ‘The alteration of the intrachain bond….’

Line 302-304 – Do you mean, ‘….elucidation of the effect the analogues have on the activation of the insulin signalling pathway?’

Line 306 – Should read, ‘….long term studies…..’

Line 307 – Should read, ‘Not only, would the studies validate the efficacy of the analogues but also could uncover any potential toxicological challenges.’

Line 324 – Should read, ‘…but was found to possess little to no blood glucose-lowering capability.’

Line 330 – Should read, ‘….when the entirety of both chains was utilised.’

Line 331 – Should read, ‘Retaining the same PEG molecule and the whole A chain resulted in an improvement in insulin activitiy…’

Line 334 – Should read, ‘…exhibits a duration of biological signalling equivalent…….’

Line 337 – Should read, ‘…retained the binding efficiency to the activated insulin receptor.’

Line 339 – Should read, ‘…similar to the short-acting insulin analogue Lispro’.

Line 341 – Should read, ‘….incubated at 45 and 75 degrees centigrade respectively.’

Line 343 – Should read, ‘…represents a breakthrough that……’

Line 345 – Should read, ‘….in areas that contend with an intermittent power supply.’

Line 346-347 – Should read, ‘….could further lessen the burden on cool insulin storage conditions while travelling.’

Line 349 – Should read, ‘..the increase in temperatures…’

Line 350 to 351 – Should read, ‘Therefore, such attempts should be encouraged and invested in and warrants further developments in studies on long-term stability, the efficacy of glycaemic profiling and toxicological profiles’.

Line 358 – You need to state what concentration of blood glucose

Line 359 – Should read, ‘…whose bioactivity is modulated by changes in blood glucose concentration…..’

Line 368 – Should read, ‘…coupling insulin with a (mono)saccharide…..’

Line 369 – Should read, ‘Affinity for the…..’

Line 373 – Should read, ‘…while a lower glucose concentration…..’

Line 377 – Should read, …..saccharide-insulin complex has been achieved through the synthesis of insulin MK-2640….’

Line 383 – Do you mean juvenile pigs (i.e. piglets) or do you mean guinea pigs?

Line 387 – Do you mean the collation of long term studies?

Line 388 – Should read, ‘within an acceptable range’,

Line 396 – Should read, ‘…solubility behaviour at various pH values.’

Lines 412 to 413 does not make sense. What are you trying to say here?

Line 455 – Should read, ‘…considering its size given the smaller fenestrations found in the glomerulus.’

Line 458 – Should read, ‘….the organ’s physiological needs are met.’

Line 460 – Should read, ‘…insulin to enter the portal vein,…..’

Line 461 – Should read, ‘…represents a strategy in which the hepatic needs for insulin could be met.’

Line 465 – Should read, ‘….currently in advanced clinical strategies.’

Line 475 – Should read, ‘The three-dimensional molecular design of the Icodec insulin analogue resembles…..’

Line 481 – Should read, ‘….which is biologically active.’

Line 485 – Should read, ‘a mitogenetic risk’.

Line 486 – Should read, ‘The Icodec insulin analogue…..’

Line 488 – Should read, ‘….a slow on-set….’

Line 495 – Should read, ‘The three-dimensional molecular design of the Etsifora insulin analogue….’

Line 509 – Are the doses one-off i.v. bolus dose injections or are they mg kg-1 administrations?

Line 513 – Should read, ‘…were observed to be 25% lower compared to the Degludec insulin analogue.’

Line 520 – Should read, ‘..it universally accessible.’

Line 520 – Should read, ‘these efforts could militate against patient non-compliance,….’

Line 525 – Should read, ‘…resulting in reduced mortality…..’

Line 527 to the end of the sentence in Line 528 does not make sense. What are you trying to say here?

In table 1 – carbon-based linker – for 60oC, there should be a space between the number and the unit (i.e. 60 oC);

Single-chain insulin (SCI) in-vitro and in-vivo should be in italics (i.e.in-vitro and in-vivo);

Line 542 – Should read, ‘…insulin therapy is fast-paced…..’

Line 545 – Should read, ‘…a greater understanding of the three-dimensional molecular structure of insulin….’

Line 550 – Should be space between the number and the unit (i.e. 2-8 oC)

Line 551 – Should read, ‘..with a hot climate and without recourse to refrigeration.’

Line 561 – Should read, ‘….in the management of diabetes….’

Line 563 – Should read, ‘…especially with the short-acting analogues.’

Line 564-565 – Should read, ‘Despite the benefit of…..’

Line 574 – Should read, ‘Lastly, the emergence of …..’

Line 577 – Should read, ‘..has a concomitant effect on the liver.’

Line 578 – Should read, ‘the goal should be to leverage the unique strengths presented by each analogue to develop….’

Line 579-580 – Should read, ‘By carefully analysing the key advantages inherent in their design and engineering strategies, scientists and clinicians can collaborate in investigating the possibility of developing more effective insulin analogues.’

Line 585 – Should read, ‘…our goal should be to strive….’

Line 599 – Should read, ‘…is associated with a complicated synthesis procedure….’

Line 617 – Should read, ‘  significant breakthroughs…..’

Line 618 – Should read, ‘…profiles…’

Line 620 – Should read, ‘…beyond the desirable emergence…..’

Line 627 – Should read, ‘…which are at the advanced stages of clinical development.’

Lines 630 to 638 should be deleted. These are part of the instructions to the author(s)

Line 691 – in-vitro should be in italics (i.e. in-vitro)

Line 707 – you need to state the date this was accessed online

Line 709 – you need to state the date this was accessed online

Line 723 – you need to state the date this was accessed online

Line 727 – you need to state the date this was accessed online

Line 772 – in vitro and in vivo need to be in italics (i.e. in-vitro and in-vivo)

Line 818 – et al should be in italics (i.e. et al.)

Line 820 – et al should be in italics (i.e. et al.)

Line 825 -  et al should be in italics (i.e. et al.)

Line 828  - et al should be in italics (i.e. et al.)

Line 833 – et al should be in italics (i.e. et al.)

Line 835 – et al should be in italics (i.e. et al.)

Line 842 – et al should be in italics (i.e. et al.)

Line 844 – et al should be in italics (i.e. et al.)

Line 846 – et al should be in italics (i.e. et al.)

Line 848 – et al should be in italics (i.e. et al.)

Line 859 – et al should be in italics (i.e. et al.)

Round 2

Reviewer 1 Report

Comments and Suggestions for Authors

I have no further comments.

Author Response

We thank the reviewer for the guidance. 

Reviewer 2 Report

Comments and Suggestions for Authors

The authors have responded positively to my previous comments (which were quite detailed), and they have addressed all of the points raised to enhance the manuscript. As well as moderating the text, the authors have included schematic representations of: the signalling transduction pathway, three dimensional renditions of insulin analogues and extra tables summarising the outcomes of the 'ONWARDS' and QWINT clinical trials. Included in the data are the number of clinically significant hypoglycemic episodes, and in the case of 'QWINT' the percentage rates of hypoglycemic events. Recognition of this data is relevant for contextualising the effect of the various insulin analogues that were deployed during the course of the trials.

Line 641 should read - ....substantially from thermostable insulin analogues.......

Line 643 to 634 should read - .....that arise from the loss of insulin potency due to suboptimal storage conditions.

Line 655 should read - ...scientists and clinicians can collaborate.....

Author Response

We thank the review for pointing our these corrections. 

We have addressed them accordingly, please see the revised manuscript